# Transcriptomic Analysis of *E. coli* after Exposure to a Sublethal Concentration of Hydrogen Peroxide Revealed a Coordinated Up-Regulation of the Cysteine Biosynthesis Pathway

**DOI:** 10.3390/antiox11040655

**Published:** 2022-03-28

**Authors:** Myriam Roth, Vincent Jaquet, Sylvain Lemeille, Eve-Julie Bonetti, Yves Cambet, Patrice François, Karl-Heinz Krause

**Affiliations:** 1Department of Pathology and Immunology, Medical School, University of Geneva, 1211 Geneva, Switzerland; vincent.jaquet@unige.ch (V.J.); sylvain.lemeille@unige.ch (S.L.); karl-heinz.krause@unige.ch (K.-H.K.); 2REaders, Assay Development & Screening Unit (READS Unit), Faculty of Medecine, University of Geneva, 1211 Geneva, Switzerland; yves.cambet@unige.ch; 3Genomic Research Laboratory, Infectious Diseases Service, University Hospitals Geneva Medical Center, Michel-Servet 1, 1211 Geneva, Switzerland; eve-julie.bonetti@unige.ch (E.-J.B.); patrice.francois@genomic.ch (P.F.)

**Keywords:** *E. coli*, oxidative stress, hydrogen peroxide, RNA-seq, *cysB*, sulfur assimilation, cysteine biosynthesis

## Abstract

Hydrogen peroxide (H_2_O_2_) is a key defense component of host-microbe interaction. However, H_2_O_2_ concentrations generated by immune cells or epithelia are usually insufficient for bacterial killing and rather modulate bacterial responses. Here, we investigated the impact of sublethal H_2_O_2_ concentration on gene expression of *E. coli* BW25113 after 10 and 60 min of exposure. RNA-seq analysis revealed that approximately 12% of bacterial genes were strongly dysregulated 10 min following exposure to 2.5 mM H_2_O_2_. H_2_O_2_ exposure led to the activation of a specific antioxidant response and a general stress response. The latter was characterized by a transient down-regulation of genes involved in general metabolism, such as nucleic acid biosynthesis and translation, with a striking and coordinated down-regulation of genes involved in ribosome formation, and a sustained up-regulation of the SOS response. We confirmed the rapid transient and specific response mediated by the transcription factor OxyR leading to up-regulation of antioxidant systems, including the catalase-encoding gene (*katG*), that rapidly degrade extracellular H_2_O_2_ and promote bacterial survival. We documented a strong and transient up-regulation of genes involved in sulfur metabolism and cysteine biosynthesis, which are under the control of the transcription factor CysB. This strong specific transcriptional response to H_2_O_2_ exposure had no apparent impact on bacterial survival, but possibly replenishes the stores of oxidized cysteine and glutathione. In summary, our results demonstrate that different stress response mechanisms are activated by H_2_O_2_ exposure and highlight the cysteine synthesis as an antioxidant response in *E. coli*.

## 1. Introduction

*Escherichia coli*, a Gram-negative rod in the family Enterobacteriaceae, is a commensal member of the normal gut flora. However, *E. coli* also represents a major source of infection. For example, *E. coli* causes 75–85% of urinary tract infections [1], and is also the most frequent pathogen responsible for bacteremia (30% of the bacteremia in Switzerland) [2]. The emergence of antibiotic resistance, especially against fluoroquinolones and 3rd and 4th generations of cephalosporins [3], as well as against beta-lactam antibiotics through the production of extended spectrum beta-lactamases (ESBLs) and carbapenemases, imply that *E. coli* is a priority target for the development of new classes of antibiotics [4].

Reactive oxygen species (ROS) are oxygen derivatives that play important roles in chemistry and biology [5]. ROS can be generated by physico-chemical interactions (e.g., radiation), by eukaryotic cells [6], where they play a role in signaling, biosynthesis and host defense, as well as by bacteria, where they might participate in microbial warfare. In many instances, the initial step in the generation of ROS is the superoxide radical anion O_2_^•−^ (generated through single electron transfer to molecular oxygen), but the latter is rapidly dismutated to the more stable and hence quantitatively predominant hydrogen peroxide (H_2_O_2_). ROS play an important role in the regulation of bacterial colonization and pathogenicity, as exemplified by the innate immune system. Phagocytes, especially neutrophils and macrophages, produce large amounts of ROS following the activation of the NADPH oxidase 2 (NOX2), a process necessary for efficient bacterial killing [7]. Loss of function of NOX2 leads to a primary immunodeficiency called chronic granulomatous disease (CGD), where patients suffer from severe and chronic infections [8]. Other ROS-producing NADPH oxidases (NOX1, DUOX1/2) are found in digestive and respiratory epithelia where they control bacterial colonization [9] and bacterial gene expression [10]. Interestingly, the regulation of bacterial colonization by ROS is not limited to enzymes of the host defense, as the production of ROS by the natural human flora limits the colonization by pathogenic bacteria. Indeed, H_2_O_2_-producing *Lactobacillus* inhibit the growth of uropathogenic *E. coli* strains [11]. Such H_2_O_2_-producing *Lactobacillus* are found in the vaginal microbiota [12,13] and in the urobiome [14,15,16]. Colonization with H_2_O_2_-producing bacteria limits the expansion of potentially pathogenic *E. coli* and therefore represents a protective factor against urinary tract infections [13,15].

While it is evident that ROS plays a role in controlling bacterial colonization and pathogenicity, the underlying mechanisms are only partially understood. For example, H_2_O_2_ is able to kill bacteria in vitro, but only at high (millimolar) concentrations. The resistance of bacteria to cytotoxic concentrations of H_2_O_2_ is explained by the fact that bacteria synthesize H_2_O_2_-scavenging enzymes such as catalases (KatG, KatE) and alkyl peroxidase (AhpC) [17]. Therefore, bactericidal H_2_O_2_ concentrations are probably not easily reached in the human body [18].

Previous studies in *E. coli* have characterized ROS-specific regulators of gene expression such as the “hydrogen peroxide-inducible genes activator” OxyR and the “superoxide response regulon” SoxRS [19]. OxyR is a transcription factor that senses H_2_O_2_ at a concentration as small as 100 nM [20]. Oxidation-sensitive cysteine residues are oxidized and form an intramolecular disulfide bridge, modifying the conformation of the protein and allowing the binding to functional DNA-binding sites and induction of the transcription of specific antioxidant genes. The last detailed study of the OxyR regulon in *E. coli* showed that OxyR regulates 38 genes in 28 transcription units while RNA-seq analysis showed that 110 dysregulated genes were differentially expressed under oxidative stress in a *oxyR* deleted strain [21]. However the sole regulation of this transcription factor cannot explain the massive gene dysregulation under H_2_O_2_ exposure [22,23]. The improvements in next-generation sequencing techniques, bioinformatics analytical tools, and genome annotations allow detailed exploration of these questions.

In this study, we performed RNA-seq analysis of *E. coli* following exposure to sublethal concentration of H_2_O_2_. We observed major gene dysregulation 10 min after the addition of H_2_O_2_, in H_2_O_2_-specific regulons as well as in genes involved in more general stress responses. We identified sulfur metabolism, including the sulfate assimilation and the cysteine biosynthesis pathways, as the most significantly up-regulated metabolic pathways at 10 min. The H_2_O_2_-enhanced expression of genes from these pathways was dependent on the transcription factor CysB. Our results revealed the importance of the sulfur metabolism under oxidative stress and the role of CysB in the immediate H_2_O_2_ response in *E. coli*.

## 2. Materials and Methods

### 2.1. Bacterial Strains and Growth Conditions

All bacterial strains used in this study appear in Table 1. *E. coli* strains were cultured at 37 °C in Luria-Bertani (LB) (Becton & Dickinson, Sparks, MD, USA) broth or on Luria-Bertani Agar (Becton & Dickinson, Sparks, MD, USA). Wild-type (WT) strain refers to BW25113.

### 2.2. Growth Curves—Exposure to H_2_O_2_

An overnight culture of *E. coli* was normalized to 1.0 Mc Farland using a Densimat (bioMérieux, Marcy-l’Étoile, France) and further diluted 1:10 in fresh LB. This bacterial culture was grown in a volume of 1 mL in a 24-well plate (ThermoFisher, Waltham, MA, USA) and incubated at 37 °C with 5 mm orbital shaking in an Infinite 200PRO plate reader (Tecan, Männedorf, Switzerland). Absorbance was measured every 10 min at an optical density of 595 nm (OD_595nm_). At early exponential phase (OD_595nm_ = 0.2), H_2_O_2_ was added from a 35% *w*/*w* (Acros Organics, Geel, Belgium) solution freshly diluted at 11-fold concentration in LB and serially diluted 1:2 in LB. A volume of 100 μL was added to the 1 mL bacterial culture (total volume per wells: 1100 μL) to reach final concentrations ranging from 80 mM to 0.078 mM H_2_O_2_.

The concentration of H_2_O_2_ stock was assessed before the experiments by using a WPA Biowave DNA spectrophotometer (Biochrom, Cambridge, UK) at 240 nm with quartz cuvette with a molar extinction coefficient of 43.6.

The Doubling Time Software v3.1.0 (http://www.doublingtime.com, accessed on 18 February 2022) was used to calculate the growth rate after H_2_O_2_ addition.

### 2.3. H_2_O_2_ Measurments by Amplex Red

Bacterial cultures of *E. coli* (see conditions above) were grown to OD_595nm_ = 0.2 and H_2_O_2_ was added at a final concentration of 2.5 mM. At indicated time points, 400 µL were taken from each well and centrifuged for 4 min at 6000× *g* to remove bacteria. The supernatant was diluted 1:100 in LB. 100 µL of each sample were transferred into a 96-well black plate with clear bottom (Corning, Corning, NY, USA) in technical triplicates. Amplex Red (Invitrogen, Carlsbad, CA, USA) was used to detect H_2_O_2_ according to manufacturer’s instructions, by the addition of 100 µL of Amplex Red mix to each well for a final concentration of 55 µM Amplex Red and 10^−2^ UI/mL Horseradish peroxidase. The plate was incubated 1 h at 37 °C and the absorbance at 575 nm was read in a Spectramax Paradigm (Molecular Devices, San Jose, CA, USA).

A H_2_O_2_ calibration curve was generated by 1:2 serial dilutions of H_2_O_2_ in sterile LB (from 0.125 mM to 2.4 × 10^−4^ mM) and used to calculate the H_2_O_2_ concentration of the samples using linear regression.

### 2.4. RNA Isolation, RNA-Seq

2 mL of bacterial cultures (see conditions above) were harvested 10 and 60 min after H_2_O_2_ addition. RNA was extracted using the TRIzol^®^ Max™ Bacterial RNA Isolation Kit (ThermoFisher, Waltham, MA, USA) according to the manufacturer’s instructions. After elution in RNA-free water, DNA digestion was performed in liquid by adding 2.5 µL DNase I and 10 µL Buffer RDD (Qiagen) in a total volume of 100 µL for 10 min at room temperature. Potential contaminants were removed by using the RNeasy mini kit plus (Qiagen, Germantown, MD, USA) according to the manufacturer’s instructions. Analysis of total RNA quality and quantity was performed using Nanodrop and Agilent 2100 BioAnalyzer (Agilent Technologies, Santa Clara, CA, USA).

For RNA sequencing, 1.2 µg of total RNA was ribo-depleted with the RiboMinus™ Bacteria 2.0 Transcriptome Isolation kit (ThermoFisher Scientific, Waltham, MA, USA). The truseq total RNA stranded kit (Illumina, San Diego, CA, USA) was then used for the library preparation. Library quantity was measured by Qubit and quality was assessed with a Tapestation on a DNA High sensitivity chip (Agilent Technologies, Santa Clara, CA, USA). Libraries were pooled at equimolarity for clustering. Single-read sequencing (100 bases) was performed using the SBS chemistry on an Illumina HiSeq 4000 sequencer. Raw data have been deposited at ENA (https://www.ebi.ac.uk/ena, accessed on 4 March 2022) under the following accession number: PRJEB51098.

The RNA-seq data were analyzed as follows. Fastq reads were mapped to the ENSEMBL reference genome (Ecolibw25113.48) using STAR (version 2.4.0j) [25] with standard settings, except that any reads mapping to more than one location in the genome (ambiguous reads) were discarded (m = 1). All reads were reported using featureCounts version 1.4.6-p1 [26]. Levels of expression were reported as raw counts and, in parallel, normalized in RPKM in order to filter out genes with low expression values (<1 RPKM) before calling genes as differentially expressed. Library size normalizations and differential gene expression calculations were performed using the package edgeR [27] designed for the R software [22]. Only genes having a significant fold-change (Benjamini-Hochberg corrected *p*-value < 0.01) were considered for the rest of the RNA-seq analysis.

Genes under the regulation (regulon) of selected transcription factors were investigated in the Ecocyc database [28] and used to generate heatmaps.

Genes were categorized in different expression patterns depending on if they were significantly up-regulated, down-regulated, or unchanged at the 2 time points, leading to 8 different groups. STRING (Search Tool for the Retrieval of Interacting Genes/Proteins) web software [29] was used to analyze physical or functional interactions in each of the 8 groups of genes.

GO term enrichment was performed using homemade scripts for the R software [30]. Gene set enrichment analysis (GSEA) was performed, without threshold of *p*-value or fold-change, using the KEGG metabolic pathways (KEGG http://www.genome.jp/kegg/, accessed on 15 December 2021) relative to Ecolibw25113.48 to generate gene sets. Genes were ranked by their calculated fold-changes (decreasing ranking). A gene set analysis using the GSEA package Version 2.2 [31,32] from the Broad Institute (MIT, Cambridge, MA, USA) was used to analyze the pattern of differential gene expression between the two groups. Gene set permutations were performed 1000 times for each analysis. The Normalized Enrichment Score (NES) was calculated for each gene set. GSEA results with a nominal FDR < 0.05 and abs(NES) > 1 were considered significant.

ISMARA analysis [33] was performed to retrieve transcription factors implicated in causing changes in gene expression. Raw fastq data were uploaded to the ISMARA web server (Swiss Institute of Bioinformatics, Basel, Switzerland). Four replicates were averaged for each condition.

### 2.5. qRT-PCR

qRT-PCR was performed on RNA samples prepared as described above. The cDNA was produced by reverse-transcribing 400 ng of total RNA using a mix of random hexamers and oligo d(T) primers and Primescript reverse transcriptase enzyme (Takara Bio, San Jose, CA, USA). The efficiency of each pair of primers was tested with serial dilutions of cDNA. Oligonucleotides are indicated in Table 2. PCR reactions (10 µL volume) contained 1:20 diluted cDNA, 2 × Power SYBR Green Master Mix (Applied Biosystems by ThermoFisher, Waltham, MA, USA), and 300 nM of forward and reverse primers. PCRs were performed on a SDS 7900 HT instrument (Applied Biosystems by ThermoFisher, Waltham, MA, USA) with the following parameters: 50 °C for two minutes, 95 °C for ten minutes, and 45 cycles of 95 °C 15 s, 60 °C one minute. Each reaction was performed in three replicates on 384-well plate. Raw Ct values obtained with SDS 2.2 (Applied Biosystems by Thermo Fisher, Waltham, MA, USA) were imported into Excel and normalization factors were calculated using the GeNorm method as described by Vandesompele et al. [34].

### 2.6. P1 Transduction

*cysB* and *katG* genes were deleted in *E. coli* BW25113 using P1 transduction from the Keio library strain as previously described [36]. The mutants were validated using PCR with appropriate gene-specific primers (Table 2).

### 2.7. H_2_O_2_ Sensivity Testing by Disk Diffusion Assay

To assess the sensitivity to H_2_O_2_, we adapted the EUCAST methods of disc diffusion assay to H_2_O_2_ testing [37]. Briefly, an overnight culture of bacteria was diluted in LB to McFarland 0.5 using a Densimat (bioMérieux Marcy-l’Étoile, France) and this was used to inoculate LB agar plates. A sterile disk of cellulose of 6 mm diameter from blotting paper was placed on the plate and 10 mL of 1 M H_2_O_2_ diluted in sterile water was added in the center of the disk. Plates were incubated at 37 °C for 18 h and the diameter of inhibition was measured.

### 2.8. Softwares

Graphpad Prism v.9.3.1 for Windows (GraphPad Software, San Diego, CA, USA) was used for data processing, graph plotting and statistical analysis. Inkscape v.1.1.1 for Windows was used for image editing (https://inkscape.org).

## 3. Results

### 3.1. Sublethal H_2_O_2_ Concentration Determination

To determine the sublethal H_2_O_2_ concentration, the WT *E. coli* BW25113 strain was grown in LB medium until an OD of 0.2 and was then exposed to a range of H_2_O_2_ concentrations (Figure 1A). H_2_O_2_ was added during the exponential phase of growth where the bacterial population is increasing and where quorum-sensing signaling that could mask gene dysregulation is minimal. We defined the sublethal concentration as the highest concentration with a similar profile to the untreated control (Figure 1B). This corresponded to 2.5 mM H_2_O_2_ and the induction of a growth lag time of approximatively 30 min.

We followed the degradation of 2.5 mM H_2_O_2_ by *E. coli* BW25113 using the H_2_O_2_-sensitive probe Amplex Red/HRP. We observed rapid and complete degradation of the exogenous H_2_O_2_ after 25 min (Figure 1C). This rapid degradation is relevant for our experiments, as results obtained 10 min after the addition of H_2_O_2_ reflected direct responses to high H_2_O_2_ concentrations, while results obtained after 60 min reflected the long-term impact of transient H_2_O_2_ exposure.

We next investigated the impact of increasing H_2_O_2_ concentrations on a *katG* deficient mutant. The bifunctional hydroperoxidase I (HPI) encoded by *katG* possesses both catalase and peroxidase. The simplified term “catalase” is further used in the manuscript. This enzyme is *E. coli*’s principal H_2_O_2_ scavenger at high H_2_O_2_ concentrations. As expected [38], the *katG* deficient mutant was at least 10 times more sensitive to H_2_O_2_ compared to wild-type (Figure 1D).

### 3.2. RNA-seq under Sublethal H_2_O_2_ Exposure

Figure 2 shows the changes in gene expression caused by H_2_O_2_ exposure after 10 min and 60 min compared to a H_2_O_2_-free condition at identical time points (Figure 2A,B). Each condition was compared to an untreated control at identical time points. A massive gene dysregulation was observed 10 min after the addition of H_2_O_2_. In order to focus on the most relevant phenomena, we used stringent criteria to determine significance (adjusted *p*-value < 0.01, Fold-change ≥ 5). With such criteria, 12.2% of all transcripts of *E. coli* were significantly dysregulated (5.3% up- and 6.8% down-regulated) in the H_2_O_2_-treated condition compared to untreated cells after 10 min, and 2.4% of transcripts were significantly dysregulated in the respective conditions at 60 min (1.5% up- and 0.9% down-regulated).

Multi-dimensional Scaling (MDS) was performed on the top 500 genes that distinguished the conditions (Figure 2C). In this analysis, distances on the plot represent the variation of expression between samples. This analysis shows that replicates of our experiments (n = 4) were near each other, ensuring a high reproducibility.

There were marked differences in gene expression after 10 and 60 min under control conditions. These differences could be explained by bacterial growth from exponential phase to mid-exponential or early stationary phase. This is supported by the fact that quorum-sensing is one of the most up-regulated pathways while flagellar assembly and ribosome formation were the most down-regulated pathways in the 60 min control condition compared to the 10 min control condition (Appendix A).

Gene expression 10 min after H_2_O_2_ exposure was highly distinct from other conditions. In contrast, at 60 min, gene expression was more similar to that of the control condition. This is most likely due to the fact that H_2_O_2_ was rapidly degraded by *E. coli* (Figure 1C). This is corroborated by results depicted in Figure 1A, showing bacterial regrowth after a lag period of approximatively 30 min after addition of 2.5 mM H_2_O_2_. In addition, time-dependent changes of media composition due to bacterial metabolism may also affect gene expression.

OxyR is a well described regulator of gene expression known to be activated by H_2_O_2_. To validate our experimental approach, we therefore investigated H_2_O_2_–induced changes in OxyR-regulated genes [28]. A total of 19 (58%) of the OxyR-up-regulated genes were found significantly up-regulated upon H_2_O_2_ exposure compared to the untreated control, while there was no marked down-regulation of OxyR-down-regulated genes. Interestingly, most of the OxyR-regulated genes returned to baseline at 60 min (Figure 2D). Thus, the activity of the OxyR regulon appears to be correlated with H_2_O_2_ concentration (Figure 1C). These results demonstrate that—as expected—the OxyR pathway was activated under our experimental conditions. However, the absence of a dysregulation of some of the OxyR genes (e.g., *dsbG*, *rscC*, *uxuA* etc.) suggests that some of these genes depend on other transcription factors or are not activated at the H_2_O_2_ concentration used in this study.

In terms of oxidative stress response, the SoxRS system, which is implicated in the defense against superoxide, redox cycling compounds and H_2_O_2,_ was also activated [19]. SoxR is activated by the oxidation of its iron-sulfur cluster and allows the transcription of the small RNA SoxS which activates the expression of approximatively 40 genes [39] (Appendix A). However, HypT, a recently discovered transcription factor activated by hypochlorite (HOCl) [40], a key ROS of the neutrophil host defense, did not induce its target genes, confirming the specificity of the OxyR and SoxRS activation by H_2_O_2_.

One of the best documented genes of the OxyR regulon is *katG*, which encodes *E. coli*’s primary H_2_O_2_ scavenger at high H_2_O_2_ concentrations. We validated *katG* induction by H_2_O_2_ by measuring its mRNA expression by qRT-PCR following treatment with increasing concentrations of H_2_O_2_ (Figure 2E). Sub-millimolar concentrations of H_2_O_2_ were sufficient for activation of *katG* gene expression and *katG* expression increased in a concentration-dependent manner until it was massively up-regulated at the sublethal concentration of 2.5 mM H_2_O_2_. These results validated the activation of OxyR under our experimental conditions, and, together with Figure 1D, confirmed that the relative H_2_O_2_ resistance of the BW25113 strain of *E. coli* is—at least in part—due to induction of the expression of *katG*.

### 3.3. Comparison between Gene Expression at 10 and 60 min after Sublethal H_2_O_2_ Treatment

In order to better define the global response to H_2_O_2_, we analyzed the time dependence of the impact of H_2_O_2_ on gene expression. Figure 3 shows the impact of H_2_O_2_ on differential gene expression after 10 min (H_2_O_2_ vs. control) compared to 60 min (H_2_O_2_ vs. control). A vast majority of genes (87%) showed absence of dysregulation (see white square of Figure 3A). Each colored square represents H_2_O_2_-dependent gene dysregulated at different time points. For example, the blue square on the left top represents genes that were down-regulated at 10 min but up-regulated after 60 min. Figure 3B provides the number of dysregulated genes at the two different time points. The most represented patterns were genes that were either up- or down-regulated at 10 min and displayed no difference at 60 min. Only 13 genes displayed a sustained up-regulation and only one gene (*carB*, implicated in the first reaction of pyrimidine biosynthesis) was down-regulated at 10 and 60 min. Selected genes from each pattern are represented in Appendix A.

### 3.4. Determination of the Functionnal Interaction of Genes Following the Same Kinetic Pattern

We determined whether each group within the Venn diagram depicted in Figure 3B included genes that shared physical or functional interactions using the STRING database [29].

Genes that were down-regulated at 10 min and whose expression then normalized after 60 min were massively enriched for genes coding for ribosomal proteins. The group of genes displaying an up-regulation at 10 min followed by no differences at 60 min represented the largest category with 180 genes. There was a significant enrichment of genes from the gene ontology response to oxidative stress, which is consistent with the OxyR regulon analysis (Figure 1D), and iron homeostasis. Genes that showed a down-regulation only at 60 min were implicated in *nitrate assimilation*. Interestingly, the genes that had sustained up-regulation were almost all implicated in the SOS-response pathway.

The fact that most changes occurred at 10 min and that genes already related to oxidative stress defense, such as the OxyR regulon depicted in Figure 1D, were up-regulated at this time point suggests that other genes following the same expression pattern can also be implicated in the response to oxidative stress.

This analysis highlights the fact that most changes occurred within minutes following H_2_O_2_ exposure and that, although it was relevant to analyze gene expression at the 60 min time point, several of the observed changes are in fact not directly due to H_2_O_2_ treatment. Thus, we mostly focused the following analyses on the direct response to H_2_O_2_ (10 min time point).

### 3.5. Identification of Transcription Factors Involved in ROS-Dependent Gene Regulation

The ISMARA (Integrated System for Motif Activity Response Analysis) software (Swiss Institute of Bioinformatics, Basel, Switzerland) was used on RNA-seq data to detect motifs in the promoters of dysregulated genes and deduce the involvement of given regulators (such as transcription factors, small RNA, RNA-polymerase subunit). Figure 4 shows 10 transcription factors that were identified as the most significant hits by the ISMARA analysis at the 10 min time point. The significance of the expression of the respective target genes at 10 min (dark purple) and 60 min (light purple) after the addition of H_2_O_2_ are illustrated. After 10 min, the target genes of 8 transcription factors were up-regulated, while the target genes of two regulators were down-regulated. At the 60 min time point, target genes of 8 of the regulators had returned to baseline levels of expression. Interestingly, the LexA target genes (63 genes distributed on 42 transcription units [28]) remained significantly up-regulated even after 60 min, while the RpoS target genes (up to 10% of the genes in *E. coli* [41]) were up-regulated after 10 min, but down-regulated after 60 min.

Note that the ISMARA results reflect the changes of targeted genes under the regulation of a given transcription factor and not necessarily its activity. For example, Fur target genes are up-regulated. However, as Fur is a transcription repressor, our results suggest that Fur is inhibited by H_2_O_2_.

### 3.6. Pathway Enrichment Analysis of H_2_O_2_—Dysregulated Genes after 10 min

As metabolism seemed highly impacted, we also performed a Gene Set Enrichment Analysis (GSEA) using the KEGG pathway database (Figure 5). Ribosome, fatty acid metabolism and oxidative phosphorylation pathways were the most down-regulated pathways. On the other side, the most up-regulated metabolic pathways were the sulfur metabolism and the siderophore biosynthesis pathways, which concurs with the up-regulation of the Fur and CysB regulated genes documented in Figure 4.

### 3.7. Investigation of the Sulfur Metabolism of E. coli under Sublethal H_2_O_2_ Concentration

The regulation of the sulfate assimilation and cysteine biosynthesis pathways have so far received little attention in the context of oxidative stress. We observed a strong up-regulation of genes regulated by CysB, the regulator of the cysteine regulon, 10 min after H_2_O_2_ exposure, which returned to baseline after 60 min (Figure 6A). Genes involved in sulfate assimilation (*cysA*, *cysW*, *cysU*, *cysP*, *spb*, *cysN*, *cysD*, *cysI*, *cysJ*, *cysH*) and cysteine biosynthesis (*cysM*, *cysK*), that are distributed in 4 distinct transcription units, showed the highest up-regulation, suggesting coordinated transcriptional regulation.

In order to test the significance of the CysB regulon in the presence of H_2_O_2,_ a single deletion mutant of *cysB* was tested in a H_2_O_2_ disk diffusion assay (Figure 6B,C). The *cysB* mutant did not exhibit an increased sensitivity toward H_2_O_2_ compared to the *katG* deleted mutant used as control. However, the *cysB* deletion led to an impaired fitness, indicated by the formation of small and translucent colonies on agar plates and a slower growth rate in liquid medium, confirming the importance of CysB for optimal growth (Appendix A).

We used qRT-PCR to investigate the expression of several CysB-regulated genes in response to increasing concentration of H_2_O_2_ (Figure 7). We observed a strong and dose-dependent up-regulation of *cysI*, *cysJ*, *cysH*, *cysN* and *tcyP*, which was completely mitigated in the *cysB* mutant at 2.5 mM H_2_O_2_. Thus, the induction of these genes is specific for H_2_O_2_ and is totally dependent on *cysB*. Note that the *cysE* gene, which is known to be independent of CysB [42], was indeed not influenced by *cysB* deletion. The *cysB* expression level did not show a H_2_O_2_ dose-dependent regulation and was similar to the control at 2.5 mM, suggesting that its activity does not depend on itself or OxyR.

Interestingly, the expression level of *katG* appeared to be up-regulated in the *cysB* deletion mutant compared to the WT after exposition to 2.5 mM H_2_O_2_.

Altogether, our study provides, for the first time, a quantitative snapshot of genes affected by H_2_O_2_ in *E. coli* and identified that the CysB regulon represents a specific response to H_2_O_2_.

## 4. Discussion

In this study, we analyzed the impact of sublethal H_2_O_2_ concentrations on gene expression in *E. coli*. Consistent with previous studies, the most important changes were the activation of the general stress responses and redox-specific stress responses. In particular, we highlighted that several transcription factors were regulating the H_2_O_2_ response. Most of these transcription factor activities were transient; however, the response to DNA damage (LexA) was maintained over a prolonged period. The main finding of this study was that sulfur metabolism was the most significantly up-regulated pathway and that H_2_O_2_ strongly and specifically induced the activation of the transcription factor CysB.

A role of ROS in the regulation of bacterial growth and survival is beyond doubt. H_2_O_2_ is the most abundant product of the phagocyte NADPH oxidase, as well as of certain H_2_O_2_ -generating *Lactobacilli*. Thus, the medical relevance of the impact of H_2_O_2_ on bacterial pathogenesis is clear. However other reactive species are also involved in host response, in particular halogenated oxidants generated by myeloperoxidase and lactoperoxidase, such as hypocyanite, hypobromite and hypochlorite [43]. The lethal toxicity of H_2_O_2_ to *E. coli* occurred at relatively high concentrations in our experimental system (concentrations ≥ 5 mM). However, alterations of *E. coli* gene expression were already detectable at 280 μM as observed by the up-regulation of *katG*, and as little as 100 nM of H_2_O_2_ has been documented to activate the redox-dependent transcription factor OxyR [20]. It has been calculated that H_2_O_2_ concentrations generated by neutrophils are in the micromolar order [44]. Thus, it is likely that H_2_O_2_ concentrations generated in vivo do not induce direct toxicity, but rather regulate gene expression.

In a previous study, using comparable experimental protocols, we found that *Staphylococcus aureus* resisted up to 10-fold higher H_2_O_2_ concentrations [45] as compared to *E. coli*. While *E. coli* has powerful defense mechanisms against H_2_O_2,_ such as the KatG catalase, the antioxidant defense is even more developed or efficient in some other bacteria. The example of *S. aureus* is particularly interesting, as despite its high ROS resistance, it is the organism that is the most predominant pathogen in CGD patients, devoid of the oxidant-generating phagocyte NADPH oxidase [8].

The analysis of the time-dependent impact of H_2_O_2_ exposure was complicated by the fact that the RNA-seq data showed massive differences in gene expression between 10 min and 60 min even under control conditions (i.e., without the addition of H_2_O_2_). The difference in gene expression between 10 min and 60 min suggested a transition toward late exponential phase (Appendix A). Thus, we principally concentrated our analysis on the changes observed at 10 min.

We used pathway enrichment analysis and the ISMARA algorithm for detection of gene regulators. ISMARA analysis fully depends on the quality of genome annotation to establish links between gene expression data and transcription factors. As *E. coli* is one of the most studied organisms and possesses the most complete genomic annotations, the results of this analysis are highly credible. As validation of the significance of this analysis, the widely characterized H_2_O_2_-sensitive transcription factors OxyR and SoxRS were indeed among the most activated transcription factors 10 min after the addition of H_2_O_2_.

Our results demonstrate that H_2_O_2_ treatment not only induced specific ROS-sensitive mechanisms but also induced responses that have previously been described for other stressors. However, the stress responses may be more activated by the H_2_O_2_-induced damage to DNA and other macromolecules than by the direct sensing of H_2_O_2_ itself. The stringent response is a well-described mechanism commonly activated by a large variety of environmental stresses, such as heat shock [46], amino acid, fatty acid or iron starvation [47,48]. This response profoundly modifies bacterial metabolism essentially through the inhibition of translation, DNA replication and nucleotide biosynthesis. The stringent response is highly conserved among bacteria and has previously been described to be induced by H_2_O_2_ [49]. It leads to a slow-growing, stress-resistant phenotype also implicated in bacterial pathogenicity (reviewed in [47]).

The group of Imlay et al. has extensively studied the response to H_2_O_2_ in *E. coli* and has described the activation of the SOS-response in response to 1 to 3 mM H_2_O_2_ [50]. The SOS-response is triggered by DNA damage such as UV-induced DNA lesions [51]. It regulates DNA damage repair but also promotes an elevated mutation rate, generating genetic diversity and adaptation, and is a key factor of antibiotic resistance emergence (reviewed in [52]). As H_2_O_2_ exposure can also cause DNA damage, it triggers the SOS-response in *E. coli* [53] and in other Gram-negative [54,55] as well as Gram-positive bacteria [56]. In our analysis, LexA, the transcription factor that regulates the SOS-response, was among the top 10 identified regulators at 10 min. It is particularly interesting that, as opposed to other stress responses, only genes from the SOS-response were up-regulated at both 10 min and 60 min. This suggests that the concentration of H_2_O_2_ used in this study led to DNA damage, which required a sustained response for efficient repair, lasting many minutes after H_2_O_2_ was fully degraded.

The alternative subunits of the RNA polymerase RpoS, the general stress response regulator [57], and RpoH that controls the heat shock response [58], have also been described to be activated by oxidative stress in *E. coli* [59,60] and were indeed among the top 10 regulators implicated after 10 min of H_2_O_2_ exposure. Deletion of these genes was previously shown to increase sensitivity to H_2_O_2_ in diverse Gram-negative bacteria, suggesting a conserved importance of their transcriptional regulation following H_2_O_2_ exposure [61,62,63]. In Gram-positive bacteria, another alternative polymerase subunit, SigB, is involved in H_2_O_2_ resistance [64,65,66]. RpoS is also known to be active during the transition between the exponential and stationary phases, or even in the mid-exponential phase [67,68], which can explain why its targeted genes appeared more highly expressed in the untreated condition at 60 min (Figure 4).

Compatible with the activation of general stress responses, we observed a massive and concerted down-regulation of genes encoding ribosomal proteins at 10 min and a transient arrest of bacterial growth for 30 min. Thus, exposure to 2.5 mM H_2_O_2_ most likely slowed bacterial protein synthesis. This is at least in part compensated by increased expression of genes relevant for the antioxidant defense. Data from the literature confirm that these genes are also up-regulated at the protein levels [38]. Similar stress responses have been shown to be activated by H_2_O_2_ in *Salmonella enterica* serovar Enteritidis [55].

Other transcriptional factors have previously been associated with the response to oxidative stress; however, the evidence is less obvious. Fur, the transcription factor regulating iron homeostasis, is up-regulated by H_2_O_2_ via OxyR activation [69,70]. As Fur acts as a repressor, it is thought to be up-regulated to inhibit excessive iron uptake that would otherwise exacerbate H_2_O_2_ damage. However, at 10 min, we observed a massive up-regulation of genes normally repressed by Fur. Other transcriptomic data of H_2_O_2_ treated *E. coli* accessible on the Gene Expression Omnibus database (GSE20305, GSE56133) and a RNA-seq experiment in H_2_O_2_ treated *Salmonella* confirmed this observation [55]. Similarly, in our analysis, the Fur-regulated siderophore biosynthesis pathway was the second most up-regulated pathway 10 min after H_2_O_2_ addition. Other studies suggest that a siderophore called enterobactin has a role in protection against H_2_O_2_ [71,72,73,74].

FNR, the transcription factor which regulates transition from aerobic to anaerobic growth appeared as one of the most altered transcription factors in our analysis. Although FNR has been suggested to sense and respond to oxidative stress [75], its mode of action has not been extensively studied in this context. In summary, the transcriptomic changes observed in our study align with the available literature and confirmed the implication of transcription factors that are less studied in the context of H_2_O_2_ exposure.

In our study, we observed a striking up-regulation of CysB regulated genes 10 min after H_2_O_2_ exposure and CysB was among the most significantly implicated regulator at 10 min. Similarly, sulfur metabolism was the most up-regulated metabolic pathway, especially genes from the sulfate assimilation and the cysteine biosynthesis pathways that were strongly up-regulated in a strikingly coordinated manner. The up-regulation of these pathways after H_2_O_2_ addition were mentioned in the literature [22,23,76], but not specifically addressed. Navigating through *E. coli* transcriptomic datasets available on the Gene Expression Omnibus database (GSE135556, GSE56133) indeed indicated an up-regulation of these pathways following H_2_O_2_ exposure. Further, the sulfur metabolism pathway is induced by H_2_O_2_ in both Gram-negative and Gram-positive bacteria, suggesting a highly conserved function [77,78].

CysB belongs to the family of LysR-type transcriptional regulators, a family of transcription factors characterized by an N-terminal helix-turn-helix (HTH) DNA binding motif and a similar amino-acid sequence [79]. This family, which contains 45 members, including OxyR, are thought to have evolved from a common ancestor due to their high homology and the conservation of their DNA-binding domain. Common features include a similar size, the formation of either homodimers or homotetramers (which is the case for CysB), the presence of the helix-turn-helix motif, and the requirement for a small molecule that acts as co-inducer. CysB is activated by sulfur limitation and the presence of N-acetyl-l-serine and leads to up-regulation of the assimilatory sulfate pathway and promotes the synthesis of the cysteine amino acid [79].

Cysteine is among the least abundant residues in proteins in both eukaryotes and prokaryotes; however, it is often highly conserved in proteins due to its reversible regulatory oxidative characteristics [80]. The biological significance of the up-regulation of cysteine biosynthesis following H_2_O_2_ is broad. Cysteine oxidation is a key regulatory element for many cellular processes such as metal binding and formation of di-sulfur bridges. However excessive oxidative stress can lead to uncontrolled and irreversible oxidation of the cysteine residues [81]. In our experimental conditions of excessive oxidative stress, irreversibly oxidized cysteine most likely needed to be renewed. Cysteine is also needed to replace the sulfur atom of H_2_O_2_-destroyed iron-sulfur clusters (Fe-S), an essential cofactor of various enzymes [82]. Diverse defense mechanisms against oxidative stress depend on cysteine; these mechanisms include the synthesis of thioredoxin and glutathione that maintain the cell redox-potential [83], and the production of hydrogen sulfide [76] and the L-cystine/L-cysteine shuttle system that protect the periplasm against oxidative damage [84].

The deletion mutant of *cysB* displayed a reduced fitness; however, its sensitivity to H_2_O_2_ was surprisingly not different from that of the WT. As the *cysB* deleted mutant is auxotrophic for cysteine, the sensitivity was assessed in media that contains amino acids (including cysteine), which may have attenuated their sensitivity to H_2_O_2_.

The mechanism leading to the up-regulation of sulfate assimilation and cysteine biosynthesis under oxidative stress is still unknown. Cysteine biosynthesis has not been described to be regulated by the prototypical H_2_O_2_ responses induced by OxyR and SoxRS [21]. H_2_O_2_-dependent up-regulation of CysB target genes was dose-dependent and the deletion of *cysB* completely abolished their up-regulation following H_2_O_2_ exposure. This confirms the specificity of CysB as a yet undescribed signaling pathway regulating sulfur metabolism under oxidative stress. In terms of mechanism, a cysteine regulator (CymR) of *Staphylococcus aureus* is known to sense oxidative stress through the oxidation of its unique cysteine residue and to up-regulate the cysteine biosynthesis pathway under H_2_O_2_ exposure [78]. Although CysB protein structure is completely different from CymR, CysB also contains a unique cysteine residue (Cys163) in its sequence. Our findings shall stimulate further molecular studies to understand the H_2_O_2_-sensing mechanism of CysB and the implication of Cys163 in this response.

## 5. Conclusions

This study aimed at improving the knowledge of how H_2_O_2_ exposure affects gene expression in *E. coli*. The typical H_2_O_2_-sensors (OxyR, SoxRS) were confirmed to regulate genes in response to H_2_O_2_ exposure and could be considered as a positive control for our experimental conditions. We observed the implication of other stress regulators, confirming the complex response triggered by exposure to H_2_O_2_ on the transcriptome of *E. coli*. The striking up-regulation of the cysteine biosynthesis pathway through the transcription factor CysB was, for the first time, highlighted as an antioxidant response in *E. coli*.

## Figures and Tables

**Figure 1 antioxidants-11-00655-f001:**
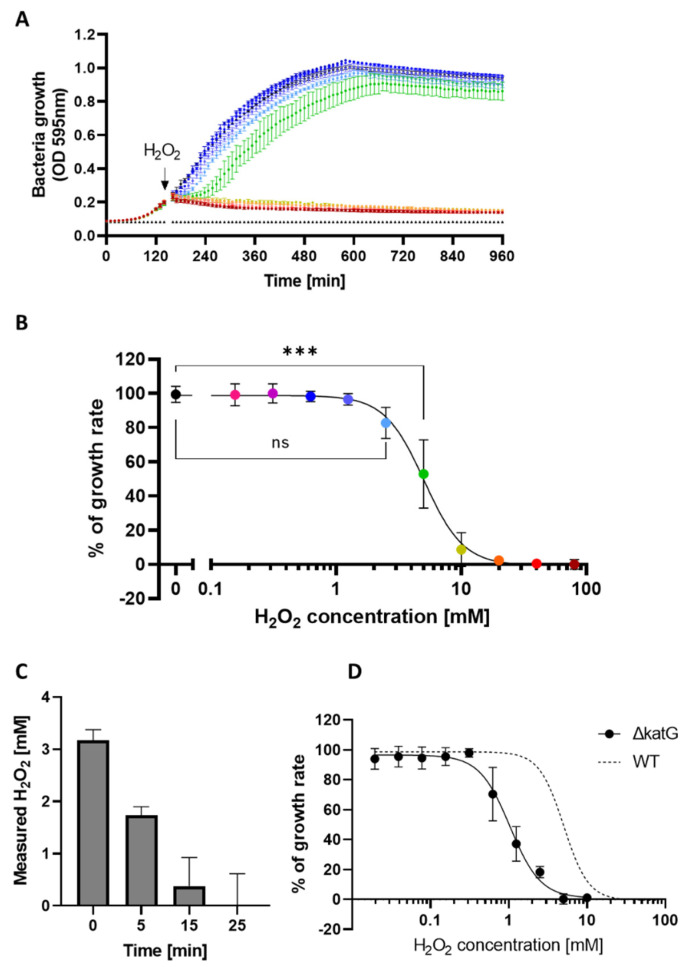
Effect of H_2_O_2_ on *E. coli* BW25113. (**A**) Bacterial growth of *E. coli* BW25113 exposed to increasing concentrations of H_2_O_2_, added during the exponential phase (OD_595nm_ = 0.2). The color code corresponds to the different H_2_O_2_ concentrations depicted in the graph in (**B**); (means of technical duplicate +/− SD; one representative experiment); (**B**) Representation of the growth rate (% of the control condition) as a function of H_2_O_2_ concentrations. The sublethal concentration was defined as the highest concentration where the growth rate was not significantly different from the control condition, namely 2.5 mM; (mean +/− SD; *N* = 3; statistical test was ANOVA with Turkey multiple comparison; *** *p* ≤ 0.001); (**C**) Quantification of H_2_O_2_ degradation using Amplex Red/HRP after addition of 2.5 mM H_2_O_2_ to bacterial culture at OD_595nm_ = 0.2 (mean +/− SD; *N* = 3); (**D**) Growth rate (% of the control condition) of the *katG* deleted mutant as a function of the H_2_O_2_ concentration, WT from panel B shown as dotted line (mean +/− SD, *N* = 3).

**Figure 2 antioxidants-11-00655-f002:**
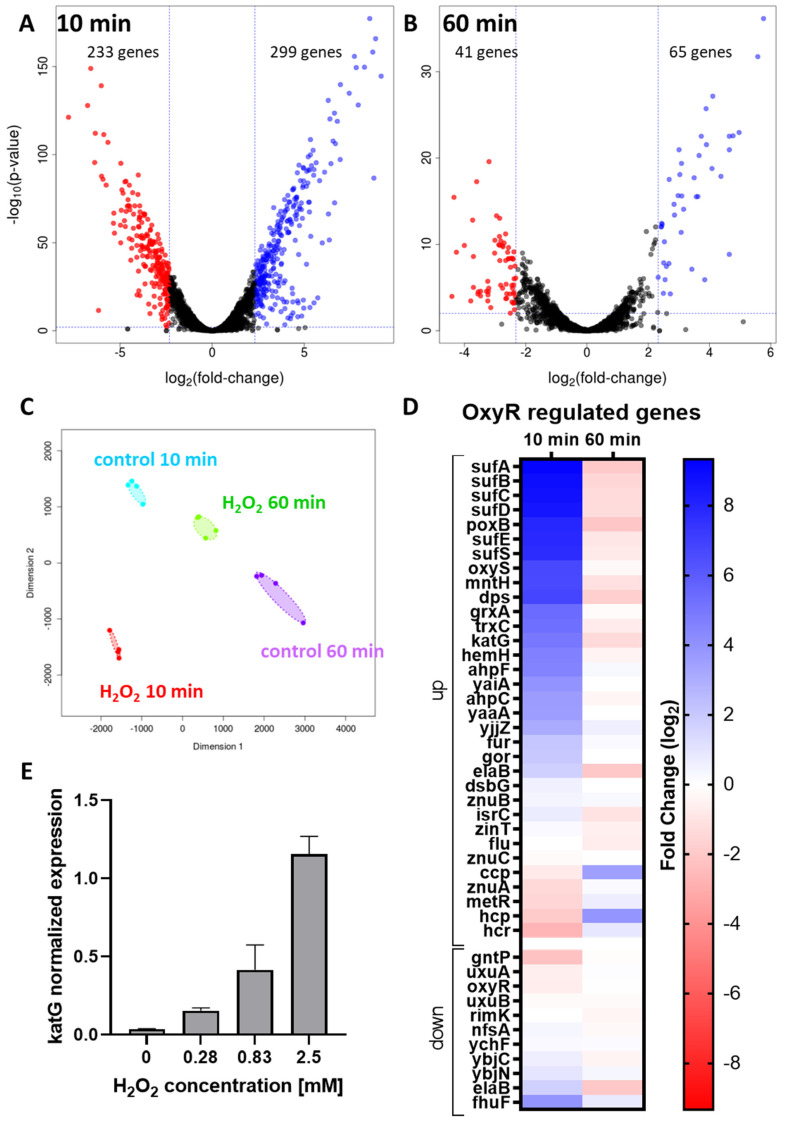
Gene expression at 10 and 60 min after exposure of *E. coli* BW25113 to 2.5 mM H_2_O_2_. Volcano plots representing the dysregulated genes at 10 (**A**) and 60 min (**B**) after H_2_O_2_ addition compared to respective untreated cells. Blue dots represent significantly up-regulated genes and red dots represent significantly down-regulated genes; thresholds were set as *p*-value = 0.01 and fold-change ≥ 5; (**C**) Multi-Dimensional Scaling plot of all samples, each color represent a condition, each dot a sample. Distances between dots represent the coefficient of variation of expression between samples for the top 500 dysregulated genes that best distinguish the samples; (**D**) Heatmap of the expression of genes of the OxyR regulon with up-regulated genes in blue and down-regulated genes in red [28] (fold change); (**E**) Catalase *katG* mRNA expression levels in response to increasing H_2_O_2_ concentrations, as measured by qRT-PCR (mean +/− SEM, *N* = 3).

**Figure 3 antioxidants-11-00655-f003:**
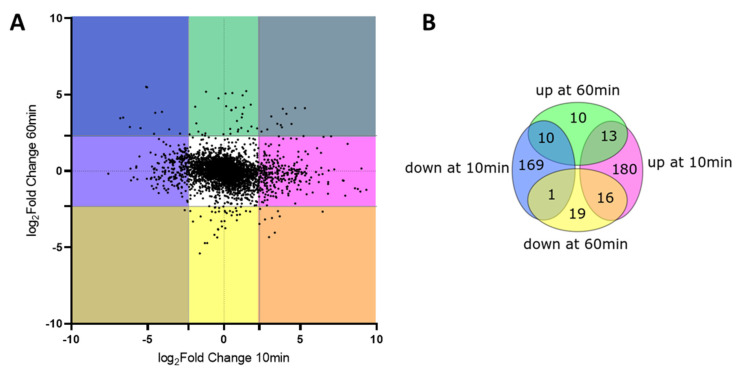
Comparison of transcriptomic changes at 10 and 60 min; (**A**) Each gene is represented by a black dot, *x*-axis: Fold Change (log_2_) at 10 min (H_2_O_2_ vs. untreated control), *y*-axis: Fold Change (log_2_) at 60 min (H_2_O_2_ vs. untreated control), pink: up-regulated at 10 min, blue: down-regulated at 10 min, green: up-regulated at 60 min, yellow: down-regulated at 60 min; (**B**) Venn diagram of conditions presented in A documenting the number of genes in each condition.

**Figure 4 antioxidants-11-00655-f004:**
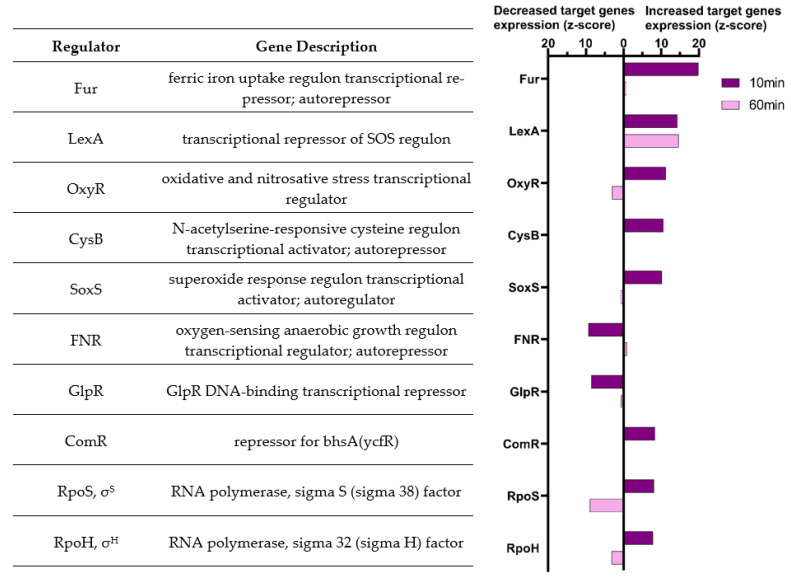
The top 10 most significant regulators implicated in gene expression changes 10 min after H_2_O_2_ exposure, using ISMARA. The significance (z-score) is illustrated for the two time points, 10 min (dark purple) and 60 min (light purple); bars extend to the left if target genes are down-regulated or to the right if they are up-regulated in the H_2_O_2_-exposed condition.

**Figure 5 antioxidants-11-00655-f005:**
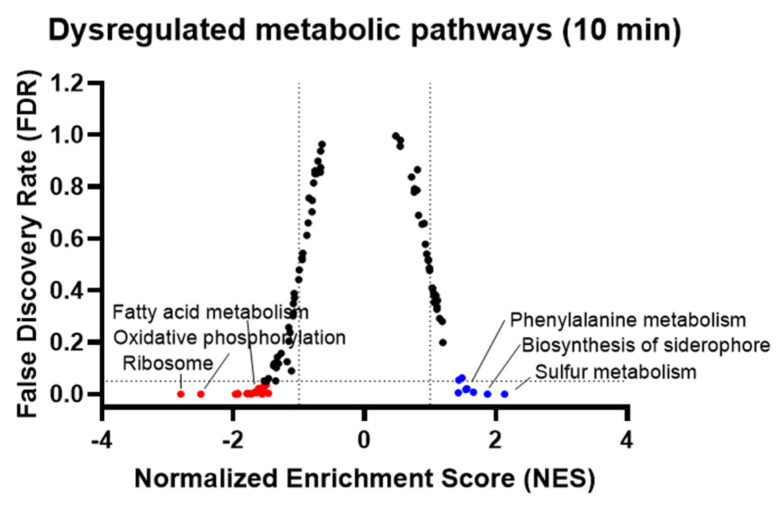
Dysregulated metabolic pathways 10 min after H_2_O_2_ addition, determined using gene set enrichment analysis. Each dot represents a metabolic pathway from the KEGG database; thresholds: FDR = 0.05, NES (normalized enrichment score) >1 and <−1.

**Figure 6 antioxidants-11-00655-f006:**
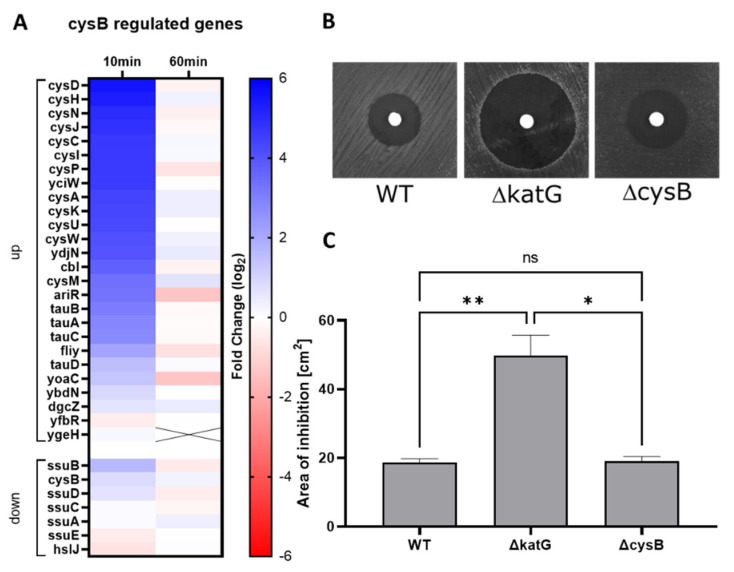
Dysregulation of the sulfur assimilation and cysteine biosynthesis pathways; (**A**) Heatmap of the differential expression of genes of the CysB regulon (fold-change (log_2_)); (**B**) Disk diffusion assay performed with 10 µL of 1 M H_2_O_2_, showing one representative experiment of the WT and the *cysB* and *katG* deletion mutants; (**C**) Zone of bacterial growth inhibition assessed by the disk diffusion assay on the WT and the *cysB* and *katG* deletion mutants (mean+/− SD; *N* = 6; statistical test was Kruskal-Wallis with Dunn multiple comparison, ns: *p* > 0.05, *: *p* ≤ 0.05, **: *p* ≤ 0.01).

**Figure 7 antioxidants-11-00655-f007:**
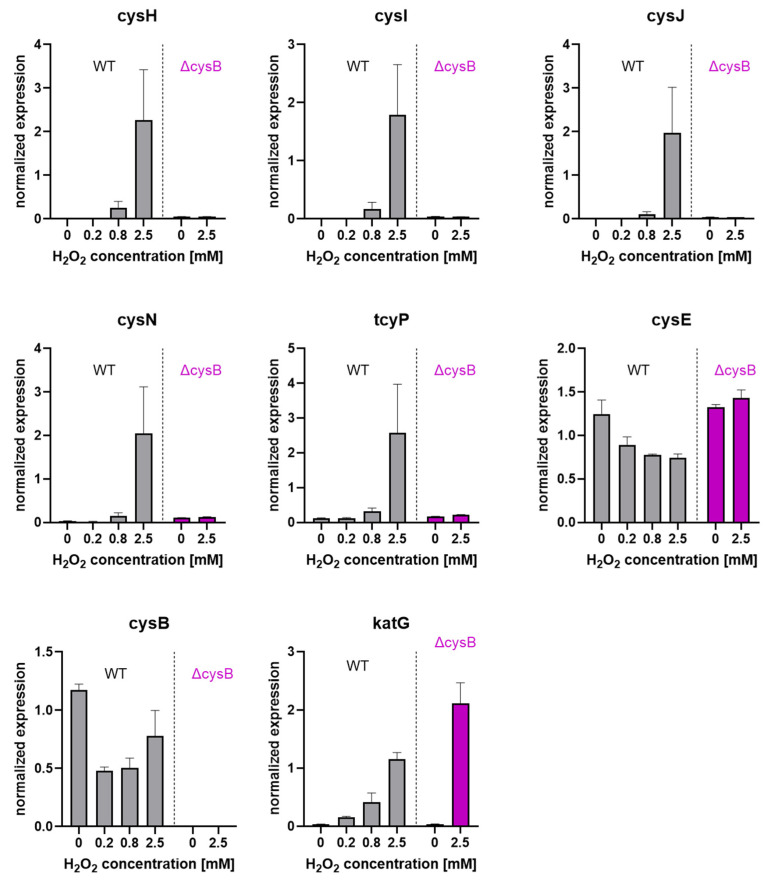
Expression of *katG* and sulfur metabolism-related genes (*cysB*, *cysE*, *cysH*, *cysI*, *cysJ*, *cysN. tcyP*) by qRT-PCR; the WT and the *cysB* deleted strain were exposed to the indicated concentrations of H_2_O_2_ (*N* = 3; mean +/− SEM).

**Table 1 antioxidants-11-00655-t001:** Strains used in this study.

Name	Genotype	Reference
BW25113	*F-Δ(araD-araB)567ΔlacZ4787(::rrnB-3)rph-1Δ(rhaDrhaB) 568hsdR514*	[24]
JW3914	BW25113, *Δkat::kan*	[24]
JW1267	BW25113, *ΔcysB::kan*	[24]

**Table 2 antioxidants-11-00655-t002:** Primers used in this study.

Name	Sequence	Efficiency (for qPCR Primers)	Reference
secA_qPCR_F	GGTAGTCGTAACGATCGCA	1.95 (95.06%)	[35]
secA_qPCR_R	TTTCCATCTCCGGTTCCAT		[35]
gyrB_N_qPCR_F	GTCCTGAAAGGGCTGGATG	1.89 (89.37%)	[35]
gyrB_N_qPCR_R	CGAATACCATGTGGTGCAGA		[35]
gyrB_V_qPCR_F	GAAATTCTCCTCCCAGACCA	1.83 (82.56%)	[35]
gyrB_V_qPCR_R	GCAGTTCGTTCATCTGCTGT		[35]
katG_qPCR_F	GGGCCGACCTGTTTATCCTC	1.92 (92.09%)	This study
katG_qPCR_R	ATCCAGATCCGGTTCCCAGA		This study
cysB_qPCR_F	ACTGTATATCGCCACCACGC	2.15 (115.12%)	This study
cysB_qPCR_R	CAGCAATTTGTGTCGGCGAG		This study
cysE_qPCR_F	CCTGCAAGGCATGAAAACCC	1.92 (91.55%)	This study
cysE_qPCR_R	CCTCTGCGGCCTGTG		This study
cysH_qPCR_F	CAGCACCGGTAAATTGGCAC	1.93 (92.60%)	This study
cysH_qPCR_R	GCACGCTACGGAAAACTGTG		This study
cysI_qPCR_F	CTTGTTGCCGTGTTCGATGG	1.93 (93.35%)	This study
cysI_qPCR_R	CGCCAACGACATGAACTTCG		This study
cysJ_qPCR_F	CTCAACGTCTTTCGCCATGC	2.03 (103.25%)	This study
cysJ_qPCR_R	GCTACGTCAAAGATGGCGTG		This study
cysN_qPCR_F	GCGACGTTTGATTCCACACC	1.95 (94.84%)	This study
cysN_qPCR_R	CTCGATTTTCGTGGTTACGCCG		This study
tcyP_qPCR_F	TCGCCGCACTGATTGTACTG	2.05 (104.82%)	This study
tcyP_qPCR_R	GCCACTAACGTTTAACGCCG		This study
katG_seq_F	ACCCTTTTTTATAAAGCATTTGTCCG		This study
katG_seq_R	GGGTTGCTCTTTCCTGCG		This study
cysB_seq_3_F	ATGTTGATGGCAAATGGGTTGAAGG		This study
cysB_seq_3_R	GCCATCACTTATCAGCAAGACG		This study

## Data Availability

Raw data have been deposited at ENA (https://www.ebi.ac.uk/ena) under the following accession number: PRJEB51098.

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
