# Peer review of "Transcriptomic Analysis of E. coli after Exposure to a Sublethal Concentration of Hydrogen Peroxide Revealed a Coordinated Up-Regulation of the Cysteine Biosynthesis Pathway"

_antioxidants, 2022, doi:10.3390/antiox11040655_

Round 1
Reviewer 1 Report
In the paper entitled "Transcriptomic analysis of E. coli after exposition to sublethal concentration of hydrogen peroxide revealed a coordinated upregulation of the cysteine biosynthesis pathway", the authors present the influence of H2O2 exposure on the gene expression of E.coli. In addition, they show an antioxidant response in E.coli through an up-regulation of cysteine biosynthesis pathway via the transcription factor CysB.
The idea of this research is a good one. Also, the paper is well organized and well written. However, it needs some minor changes before acceptance in Antioxidants:
- minor English language editing (there are some too long phrases that must be split into much easier to read and follow sentences; eg. "H2O2 exposure led to the activation of specific antioxidant response and of general stress response characterized by a transient downregulation of genes involved in general metabolism such as nucleic acid biosynthesis and translation with a striking and coordinated downregulation of genes involved in ribosome formation, and a sustained upregulation of the SOS response";
- "upregulation" or "up-regulation" (same goes for down)...I'd choose the later;
- "Gram positive" or "Gram-pozitive" (same goes for negative)...I'd choose the later;
- the phrase "against fluoroquinolones and 3rd and 4th generations (against drugs...correct) of cephalosporins [3] as well as extended spectrum beta-lactamases (ESBLs) and carbapenemases producers (against microorganisms...incorect) imply that E.coli is a priority target for the development of new classes of antibiotics"....does not make sense and must be corrected;
- page 2 - line 49: replace "(generated through single electron transfer to oxygen" with "(generated through single electron transfer to molecular oxygen";
- "in vitro" and "in vivo" should be Italic faced;
- the authors should use the same Font Type, especially within the text (tables and figures too);
- deleted unnecessary space before "3.5. Identification of transcription factors involved in ROS-dependent gene regulation";
- concerning figure 4, "decreased target genes expession (z-score)" is inside a text box that covers something still visible....also, the Font Type is different.
Author Response
In the paper entitled "Transcriptomic analysis of E. coli after exposition to sublethal concentration of hydrogen peroxide revealed a coordinated upregulation of the cysteine biosynthesis pathway", the authors present the influence of H2O2 exposure on the gene expression of E.coli. In addition, they show an antioxidant response in E.coli through an up-regulation of cysteine biosynthesis pathway via the transcription factor CysB.
The idea of this research is a good one. Also, the paper is well organized and well written. However, it needs some minor changes before acceptance in Antioxidants:
AU: The authors would like to thank reviewer 1 for the positive comment on our study and its attentive reading of the manuscript.
minor English language editing (there are some too long phrases that must be split into much easier to read and follow sentences; eg. "H2O2 exposure led to the activation of specific antioxidant response and of general stress response characterized by a transient downregulation of genes involved in general metabolism such as nucleic acid biosynthesis and translation with a striking and coordinated downregulation of genes involved in ribosome formation, and a sustained upregulation of the SOS response";
AU: The language of the manuscript was corrected by a native English speaker.
"upregulation" or "up-regulation" (same goes for down)...I'd choose the later;
"Gram positive" or "Gram-pozitive" (same goes for negative)...I'd choose the later;
AU: Modified accordingly in the revised version of the MS
the phrase "against fluoroquinolones and 3rd and 4th generations (against drugs...correct) of cephalosporins [3] as well as extended spectrum beta-lactamases (ESBLs) and carbapenemases producers (against microorganisms...incorect) imply that E.coli is a priority target for the development of new classes of antibiotics"....does not make sense and must be corrected;
AU: The sentence has been modified in the revised MS
page 2 - line 49: replace "(generated through single electron transfer to oxygen" with "(generated through single electron transfer to molecular oxygen";
AU: The replacement was done according to the suggestion.
"in vitro" and "in vivo" should be Italic faced;
AU: Modified accordingly in the revised version of the MS
the authors should use the same Font Type, especially within the text (tables and figures too);
deleted unnecessary space before "3.5. Identification of transcription factors involved in ROS-dependent gene regulation";
concerning figure 4, "decreased target genes expession (z-score)" is inside a text box that covers something still visible....also, the Font Type is different.
AU: Modified accordingly in the revised version of the manuscript
Reviewer 2 Report
Dear Authors
a very well prepared study; I think carfully proof reading should be done. See my comments below. No detailed changes requested from me.
minor points:
- E. coli; here sometimes the space is missing; check and correct.
- some small typos/space issues should be checked.
- some paragrahps end with empty lines
- The discussion is very detailed.
Author Response
Dear Authors
a very well prepared study; I think carfully proof reading should be done. See my comments below. No detailed changes requested from me.
Authors would like to thank reviewer 2 for the positive comments about our work.
minor points:
- coli; here sometimes the space is missing; check and correct.
some small typos/space issues should be checked.
some paragrahps end with empty lines
The discussion is very detailed.
AU: All suggestions have been implemented in the revised version of the manuscript and the language was corrected by a native English speaker.